# Design and Development of Hybrid Al_2_O_3_ Based Composites with Toughening and Self-Lubricating Second-Phase Inclusions

**DOI:** 10.3390/ma12152378

**Published:** 2019-07-25

**Authors:** Syed Sohail Akhtar, Taha Waqar, Abbas Saeed Hakeem, Abul Fazal M. Arif, Khaled Saleh Al-Athel

**Affiliations:** 1Mechanical Engineering Department, King Fahd University of Petroleum and Minerals, Dhahran 31261, Saudi Arabia; 2Center of Excellence in Nanotechnology, King Fahd University of Petroleum and Minerals, Dhahran 31261, Saudi Arabia; 3Mechanical Engineering Department, Prince Mohammad bin Fahd University, Al-Khobar 31952, Saudi Arabia; 4Mechanical Engineering Department, McMaster Manufacturing Research Institute, McMaster University, Hamilton, ON L8S 4L8, Canada

**Keywords:** ceramic, design, composites, fracture toughness, experimental, self-lubrication

## Abstract

Polycrystalline ceramics, such as alumina (Al_2_O_3_), are brittle and they generally wear by fracture mechanism, which limits their potential in tribological applications. In the present work, computational design tools are used to develop hybrid Al_2_O_3_ composites reinforced with best combinations of toughening and self-lubricating second-phase particles for cutting tool inserts in dry machining applications. A mean-field homogenization approach and J-integral based fracture toughness models are employed to predict the effective structural properties (such as elastic modulus and fracture toughness) and related to the intrinsic attributes of second- phase inclusions in Al_2_O_3_ matrix. Silicon carbide (SiC), boron nitride (cBN and hBN), zirconia (ZrO_2_), graphite, titanium dioxide (TiO_2_), and titanium carbide (TiC) were found the most suitable candidates to be added in Al_2_O_3_ matrix as individual or hybrid combinations. A series of samples including standalone Al_2_O_3_, single inclusion composites (Al_2_O_3_/SiC, Al_2_O_3_/cBN) and hybrid composites (Al_2_O_3_/SiC/cBN, Al_2_O_3_/SiC/TiO_2_ and Al_2_O_3_/SiC/graphite) are sintered by Spark Plasma Sintering (SPS) for validation purpose. Properties of the sintered composites are measured and compared with the proposed computational material design. Composition and phase transformation of the sintered samples are studied using X-ray diffraction (XRD) and Raman spectroscopy, while their morphology is studied using Field Emission Scanning Electron Microscope (FESEM). The presented nontraditional material design approach is found to significantly reduce experimental time and cost of materials in developing toughened and anti-friction ceramic composites.

## 1. Introduction

The global drive of innovation has led to the manufacturing of complex high-quality products in various technological areas. This in turn has raised the need for cost efficient production which has raised research interest within the industry for enhanced cutting tool material design [1]. Reducing the use of coolant can reduce cost of machining process while being environmentally friendly. As dry cutting becomes an increasingly favored process, this requires cutting tools to maintain structural properties such as fracture toughness at high temperatures [2,3]. Cutting inserts provides a relatively cheaper alternative to regular cutting tools as only the insert needs to be replaced which saves money and time. Materials commonly considered as inserts for hard work pieces include ceramics (such as Al_2_O_3_, Si_3_N_4_ etc.), cemented carbides, cBN and polycrystalline diamond. For a tool insert to be at its most effective, it would require excellent fracture toughness properties, good wear resistance, thermal and mechanical shock resistance, and tribological attributes.

Some of the most commonly used ceramic-based cutting tool materials include Al_2_O_3_ and silicon nitride (Si_3_N_4_). One of the key advantages of ceramic-based tools is their potential to keep their structural integrity at elevated temperatures [2,3,4,5]. For example, ceramic-based tools become ductile at 2200 °C in contrast to carbide tools, which soften at around the 870 °C region [2]. Therefore, ceramics allow machining to be carried out at high temperatures. This is beneficial when considering the machining of newly introduced super hard materials. Development of ceramic-based cutting tools has greatly risen during recent times and they are starting to be considered over carbide tools in different high-speed machining applications. Furthermore, ceramics are deemed superior in terms of structural properties at high temperatures compared to those of carbides. They maintain excellent hardness and are chemically stable at temperatures above 1000 °C. Nevertheless, they possess several deficiencies such as low fracture toughness, low mechanical and thermal shock resistance, and an affinity to fail via chipping. Therefore, there is a need to improve certain properties such as elastic modulus and fracture toughness to enhance structural performance of ceramic-based cutting tool materials. In recent published works [6,7], there has been considerable progress in designing new ceramic-based composites reinforced with whisker particles such as SiC. Particular focus has been on the improvement of key structural properties such as fracture toughness for high temperature machining processes [1]. With the inclusion of multiple phases, it is possible to enhance the targeted “field properties”, which include fracture toughness, elastic modulus and coefficient of thermal expansion, and therefore improve the fracture behavior of the developed composite as well as tribological characteristics. The tool life and overall performance of the cutting tool inserts directly depend on these properties. It has been shown that the Al_2_O_3_ reinforced with SiC whiskers tends to extend the tool life, which is attributed to the enhanced toughness due to the presence of a hard secondary phase [6,7,8]. It has been reported that addition of 20% SiC in Al_2_O_3_ tends to retain the flexural strength even when the composite is exposed to an elevated temperature of around 900 °C. Xu et al. [9] developed Al_2_O_3_-TiC composites and reported an enhanced fracture toughness of 5.89 MPa·m^1/2^ which is approximately 20% higher than that of the corresponding ceramic material containing no TiC. It has been reported [10] that the incorporation of a ductile phase into a brittle ceramic composite can result in the improvement of toughness, which is due to the dissipation of energy associated with crack propagation. Al_2_O_3_ reinforced with Ti [C, N] fabricated using a newly introduced method of repetitious hot-pressing exhibited enhanced structural properties with a decreased coefficient of friction reported [11,12]. Kumar et al. [13] reported Al_2_O_3_ based composites that were reinforced with ceria that demonstrated an improved hardness as well as an enhancement in fracture toughness and hardness in comparison to standalone Al_2_O_3_ and on par with zirconia toughened alumina (ZTA). Al_2_O_3_ mixed with 30% cBN ceramic composites were manufactured through spark plasma sintering, which exhibited enhanced structural properties [14]. Jianxin et al. [15] manufactured TiB_2_ reinforced Al_2_O_3_ matrix composite for cutting tools using hot pressing technique. They reported an improvement in the wear resistance due to the formation of tribo film during cutting action, which was found to offer reduction in costs in machining processes and reduce the impact on the environment due to a reduced need for coolant. They extended their research and developed hybrid composite by adding SiC as second inclusion in addition to TiB_2_. They found that the structural properties were enhanced by incorporating additional SiC content [16]. Bronizewski et al. [17] developed Al_2_O_3_ composites reinforced with graphene oxide using spark plasma sintering process and reported a significant cohesion between the Al_2_O_3_ and graphene oxide interface. S. Jahanmir and X. Dong [18] conducted a detailed wear analysis of Al_2_O_3_ and found that mechanisms such as tribo-chemical reaction, micro-fracture, plastic deformation and plowing were the major mechanisms in affecting the overall tribological characteristics. Hence, the performance of standalone Al_2_O_3_ can be enhanced if its fracture toughness could be enhanced and coefficient of friction could be reduced, which is possible by reinforcing Al_2_O_3_ with the best combinations of toughening and self-lubricating particles. Tribological characteristics and mechanism of wear of Al_2_O_3_ were studied by S. Jahanmir and X. Dong [19]. Kopeliovich [20] reported that the coefficient of friction of Al_2_O_3_ surface could be reduced by self-lubricating particles such as graphite, hBN, metal sulphides and metal oxides.

It is important to mention that nearly all surveyed literature regarding development of ceramic composites has been carried out using purely an experimental approach. Using a computational material design approach is found advantageous in many aspects, which includes reduction in resources, material consumption during repetitive experiments together with time required to conduct these experiments. In this paper, a systematic material design approach is used to design enhanced Al_2_O_3_ based composites with enhanced fracture toughness and tribological attributes. For means of validation, a series of composites with designed combinations of toughening and self-lubricating particles in Al_2_O_3_ matrix are then synthesized using Spark Plasma Sintering (SPS). The prepared composites are then characterized and tested to determine the application targeted properties.

## 2. Mathematical Models Used at Material Design Stage

In the following Sections, a brief description of fracture toughness and mean field homogenization is provided.

### 2.1. Fracture Toughness Model

The effective fracture toughness of reinforced ceramic composites depends on multiple parameters which include type of second-phase inclusions and their distribution, fracture resistance of matrix and second-phase inclusions, interface fracture resistance and overall microstructure of the composite. All these parameters control the resulting fracture mechanisms such as particle cracking, matrix cracking and interface debonding. Li and Zhou [21,22] previously proposed analytical models through the use of a J-integral based method and cohesive finite element method (CFEM) to predict the overall fracture toughness of a Al_2_O_3_ composite reinforced with single TiB_2_ inclusions. They evaluated the single-phase composite as a function of inclusion size, its content and the intrinsic and constituent properties. In the current paper, their model is extended to account for hybrid composites, which can handle more than single second-phase particles. Due to statistical measures, this approach is applied on microstructures possessing random heterogeneous phase distributions. The fracture toughness (*K_IC_*) of brittle material such as Al_2_O_3_ can be correlated to critical energy release rate function (*J_IC_*) using Equation (1):(1)$$K_{IC}^{2}=J_{IC}\frac{\bar{E}}{1-{\bar{v}}^{2}}$$
where $\bar{E}$, $\bar{v}$
$,\;J_{IC}$ are the effective elastic modulus of the composite, Poisson’s ratio of the composites, and critical energy release rate function, respectively. Considering three major fracture modes, namely interface-debonding, matrix-cracking and particle-cracking, an average energy release rate function can be stated using Equation (2):(2)$$J_{IC}=\;\xi \left(Q,\;s,f\right)\left(\varphi_{in}H_{in}+\varphi_{m}H_{m}+\varphi_{p}H_{p}\right)$$

In Equation (2), *ξ* (*Q, s, f)* is defined as the crack length multiplication factor (CLMF). This factor takes care of the effect of interfacial compliance as a function of strength ratio *Q*, inclusion particle size *s* and the inclusion volume fraction *f*, respectively. *H_in_*, *H_m_*, and *H_p_* are the proportions of crack lengths that are associated with the interface-debonding, matrix-cracking and particle-cracking. *Φ_in_, Φ_m_, and Φ_p_* are the fracture energies of the interface, matrix and inclusion particle; respectively. The calculations based on Equations (1) and (2) are based on the assumption of quasi-static crack growth where the speed of the crack approaches zero. Nevertheless, it is found that for ceramics which are considered brittle materials, *J_IC_* can be determined performed by accounting for the crack speed via the use of dynamic calculations. As reported [21], there are two kinds of possible fracture types that can occur when a crack approaches an interface between matrix and inclusion within a ceramic composite. Interface debonding is a key mechanism for crack deflection, which is a result of weak interfacial-bonding, smaller inclusion particle size and greater roundness of particles. Second mode of fracture is particle cracking which is promoted by crack penetration. Energy-based criteria is used which takes into account the contribution of each fracture type along different crack paths. A crack will grow when there is sufficient energy in the stress field that is needed to form new fracture areas. Therefore, in order for crack growth, the energy release rate of each type of fracture mode would have to exceed the interface fracture energy. Characterization of the competition between the two fracture types (crack deflection and crack penetration) is done to quantify the proportion of each mode occurring.

We can predict *K_IC_* by quantifying the probability of occurrence for each fracture mechanism over a defined crack length *D* [23,24,25]. It is important to note that the matrix cracking or interfacial debonding will tend to deflect the crack. Consequently, the interfacial fracture resistance and bonding strength should be taken into account. The relation that accounts for these factors is empirically determined using Equations (3), (4) and (5):(3)$$H_{in}=\frac{\int_{0}^{D}P_{ij}\left(D\right)dx}{D}p{\left(\frac{\varphi_{in}}{\varphi_{m}}\right)}^{-a\ln Q}$$

(4)$$H_{p}=\left(\frac{\int_{0}^{D}P_{ij}\left(D\right)dx}{D}\left(1-p\right)\;+\;\frac{\int_{0}^{D}P_{jj}\left(D\right)dx}{D}\right){\left(\frac{\varphi_{in}}{\varphi_{m}}\right)}^{-b\ln Q}$$

(5)$$H_{m}=1-H_{in}-H_{p}$$

The dimensionless parameters and their values, *a* = 0.4 and *b* = 2, have been predetermined by fitting relations to CFEM results. *P_ij_* and *P_jj_* are the probabilistic functions for randomly located vectors with a defined length *D* that begins in each different phase present. Here, phase *i* is defined as the matrix with *j* considered as the inclusion of volume fractions. Additional inclusions are included by manipulating the probability functions to consider the probability of encountering these inclusions when a crack is propagated. p is the probability of crack deflection and *Q* is the strength of interfacial bond. This bond is determined as the ratio between the interfacial cohesive strength and the average value of the cohesive strengths of matrix and second-phase. *Q* may be varied between 10^−5^ and 10 based on the strength of the interface bonds (where small *Q* represents ductile interfaces).

As previously mentioned, prediction of fracture toughness *K_IC_* requires evaluation of the crack multiplication factor *ξ* (*Q, s, f*). This factor depends on variation in trajectory of heterogeneity-induced cracks which requires *ξ* > 1 for all cases. In the current work, a crack length multiplication factor considering, *ξ* at *Q* = 1 is assumed which can be presented using Equation (6):(6)$$\xi \left(s,f\right)=\frac{1-e^{D_{2}}}{e^{D_{1}}-e^{D_{2}}}e^{D_{1}f}+\frac{e^{D_{1}}-1}{e^{D_{1}}-e^{D_{2}}}e^{D_{2}f}$$
where *D_1_* and *D*_2_ are functions of inclusion particle sizes.

### 2.2. Mean-Field Homogenization Scheme

To predict the effective elastic modulus of the particle reinforced composites, the seminal work done by Eshelby in 1965, along with using Mori-Tanaka mean-field homogenization theory is used. The strain distribution at the micro scale is related to the macro scale strains using the Equation (7) of strain localization:(7)$$E\left(x\right)=A\left(x\right){\bar{E}}_{M}$$

The homogenized elasticity tensor can be solved using the Equation (8):(8)$${\bar{C}}_{M}=\underset{i}{\sum }\emptyset_{i}C_{i}A_{i}$$

A localized strain tensor of inclusion of type *i*, *A_(MT),i_*, is determined using Mori-Tanaka scheme using Equations (9) and (10):(9)$$A_{\left(MT\right),i}={\left[f_{i}I+\emptyset_{m}{\left(A_{i}\right)}^{-1}+\underset{j}{\sum }f_{i}A_{j}{\left(A_{i}\right)}^{-1}\right]}^{-1}$$

(10)$${\bar{C}}_{\left(MT\right)}=C_{M}+\underset{i}{\sum }f_{i}\left(C_{i}-C_{M}\right)A_{\left(MT\right),i}$$

Here *C_i_* denotes the elastic strain tensor of inclusions and *f_i_* the volume fraction of inclusions. The sybmol *I* is the fourth order identity. Equation (10) is used to predict the average strain field in inclusion *i* using the macroscopic strain field.

It has been reported by Doghri and Tinel [26] that predicting structural properties for more than single inclusion in the matrix using Mori-Tanaka scheme leads to some erroneous results. In the current work, a modified approach is used where the composite is divided into pseudo grains each containing a single kind of inclusion followed by mean-filed homogenization performed for each grain using Voigt method (Equation (11)). The resulting properties are predicted as a function inclusion content.

(11)$${\bar{C}}_{eff}=\overset{N}{\underset{i=1}{\sum }}\frac{\emptyset_{i}^{i}}{1-\emptyset_{m}}C_{eff}^{i}$$

Here, ${\bar{C}}_{eff}$ is the effective elasticity tensor.

### 2.3. Consideration of Second-Phase Inclusions

Al_2_O_3_ is a common material candidate for ceramic based cutting tool inserts because of its enhanced structural properties and low cost [27]. However, its relatively inferior fracture toughness limits its applications. Structural properties such as fracture toughness and tribological attributes of Al_2_O_3_ composites could be adjusted for a particular application if proper combination of second-phase particle(s) with desired sizes and volume factions are added in the matrix. As a first step, various particle inclusions are identified keeping in view their structural and tribological attributes and effective properties are predicted through in-house built codes. Some potential second-phase inclusions, as shown in Table 1, are selected based on extensive literature surveys keeping in view individual fracture resistance values and solid lubricity as major attributes. Considering the base properties of Al_2_O_3_ as a benchmark, combinations of second-phase particles are chosen such that the resulting composite properties are in the desired range. Some inclusions have been eliminated from the list due to their negative impact on the targeted effective properties. For example, diamond has been excluded because of its high brittleness and incompatibility with ferrous work pieces at high temperatures. As a preliminary step, the effect of each individual particle as a function of volume content and size is considered at the material design stage to validate the material elimination process by considering their intrinsic properties and other characteristics such as chemical affinity and tribological feature during cutting process.

## 3. Experimental Setup

Al_2_O_3_ is selected as the base matrix material for our composite design. After predictions, various second-phase particles are identified which were then procured in desired sizes. Two set of samples using two types of α-Al_2_O_3_ powders with ~15 µm and ~150 nm sizes supplied by Sigma-Aldrich (St. Louis, MO, USA) and Chempur (Karlsruhe, Germany), respectively, are used as the matrix powder for sintering the composites using SPS in order to see the effect of particle size on the microstructure. The SiC powder of size ~40 µm (Buehler, Lake Bluff, IL, USA), cBN of size ~40 µm (Element Six, ABN800, Orem, UT, USA), graphite flakes, (99% carbon) of size ~60–44 µm (provide by Sigma-Aldrich) and TiO_2_ of size around ~1 µm (by Chempur) are used as second-phase inclusions. These second-phase inclusion(s) were properly mixed into the Al_2_O_3_ matrix using ultrasonic probe sonication in ethanol solution in order to obtain greater dispersion of the particles before microscopic examination followed by overnight placement in a furnace set at 100 °C to eliminate any ethanol residue. The designed composite systems such as Al_2_O_3_/SiC, Al2O_3_/cBN, Al_2_O_3_/SiC/cBN, Al_2_O_3_/SiC/TiO_2_, and Al_2_O_3_/SiC/Graphite were prepared in a disc shape with a diameter of 20 mm and thickness of 6 mm. These composites were sintered using spark sintering system supplied by FCT (Frankenblick, Germany), which is equipped with graphite die. The sintering parameters used were sintering temperature = 1400 °C, die pressure = 50 MPa, heating rate = 100 °C/min, and holding time = 10 minutes. Since the SPS parameters have a significant effect on the resulting quality of composites, an extensive literature survey [14,28,29] is conducted to determine these parameters. The sintering parameters were kept constant in all experiments for consistency. A pyrometer placed at the bottom of the central hole located in the upper punch is used to measure and monitor sintering temperature. Standard procedure of grinding, polishing and gold coating is used to prepare the samples for microscopic analysis. A Field Emission Scanning Electron Microscope (FESEM, Lyra 3, Tescan, Brno, Czech Republic) was used to conduct the morphological analysis of the samples. Energy dispersive x-ray spectroscopy (EDX, Oxford Instruments, Abingdon, UK) attached to the FESEM was used to conduct the compositional analysis of the samples. To detect various phases, X-ray diffraction analysis was conducted using Rigaku desktop X-ray powder diffractometer model (Miniflex II, Tokyo, Japan) with Cu K_α_1 radiation and a wavelength of 0.154 nm. A load of 20 N (2000 g) was used to measure the hardness using Vicker’s hardness tester (Buehler) and to measure the fracture toughness by employing the mathematical relationship [30] using the crack length generated from the corners of the indentations. Porosity in composites is measured using theoretical density based on rule of mixture and measured density based on Archimedes principle (details can be found in authors’ previous works [29,31]). The modulus of elasticity was measured using CSM instruments Micro Combi (Needham, MA, USA) that utilizes a pyramid diamond indenter driven down into the surface of the sample. The procedure involves the loading of the indenter to a preset value followed by gradually decreasing the load until material relaxation. The slop of the tangent to the loading curve is then used to measure the modulus of elasticity of the material $\left(\;E_{IT}\right)$ using the Equation (12):(12)$$E_{IT}=\frac{1-v_{s}{}^{2}}{\frac{1}{E_{r}}-\frac{v_{i}{}^{2}}{E_{i}}}$$
where *ν_s_* is the Poisson’s ratio of the sample, *E_i_* and *ν_i_* are the elastic modulus (1141 GPa) and poison’s ratio (0.07) of the diamond indenter respectively, and the reduced modulus (*E_r_*) is calculated using data from the indentation from Equation (13):(13)$$E_{r}=\frac{\sqrt{\pi }\cdot S}{2\cdot b\cdot \sqrt{A_{P}\left(h_{c}\right)}}$$
where *S* is the stiffness or the slope of the unloading curve, *b* is a compliance constant, *h_c_* is the contact depth and *A_p_* is the projected area of contact.

## 4. Results and Discussion

The results of the computational simulations are first discussed for single-inclusions followed by hybrid-inclusion Al_2_O_3_ composites. The results are presented and analyzed as a function of second-phase inclusion type, volume fraction, size and its influence on the effective properties of resulting composite. Intrinsic properties of Al_2_O_3_ and individual second-phase particle has direct bearing on achieving the target properties of the composites, which is either determined from experiments or obtained from literature. The major focus is on the key structural target properties namely, elastic modulus and fracture toughness, which is considered as a constraint on the design process with self-lubrication characteristics. A schematic of various steps involved in material design stage are shown in Figure 1.

### 4.1. Correlation between Fracture Toughness and Single Second-Phase Inclusion in Al_2_O_3_ Matrix

Enhanced fracture toughness can make the cutting tool insert more durable and resistant to cracking at high cutting speeds, resulting in greater tool life. The computational models are first run for single inclusion in order to identify the most suitable reinforcements that are likely to enhance effective fracture toughness of the resulting composite with a potential to be used in dry cutting conditions due to self-lubrication attribute. Starting with elastic modulus as an essential property that directly affects the material stiffness and fracture toughness of resulting composite, predictions are first run for single inclusions followed by multiple phases. Figure 2a and Figure 2b, respectively, show the effective elastic modulus and fracture toughness as a function of volume fraction of various second-phase inclusions in the Al_2_O_3_ matrix. It can be observed that elastic modulus of standalone Al_2_O_3_ (measured as ~280 GPa) and its fracture toughness (predicted and measured as ~4.09 MPa·m^1/2^) are set as a benchmark in order to compare the selected inclusions. As depicted in Figure 2a, among all inclusions, graphite tends to significantly decrease the elastic modulus as its content is increased. However, its controlled amount mixed in Al_2_O_3_ matrix could be helpful in maintaining the overall stiffness of the composites while inducing self-lubricity that is needed during dry cutting application. More importantly, the introduction of graphite in Al_2_O_3_ matrix is predicted to decrease the effective fracture toughness despite its high fracture resistance (fracture energy ~85 J/m^2^) as shown in Figure 2b. This is associated with the low stiffness (27.6 GPa) of the graphite, which considerably reduces fracture toughness owing to the fact that elastic modulus is directly proportional to the square of *K_IC_*. As shown in Figure 2a, apart from Partially Stabilized Zirconia (PSZ), which slightly reduces the effective elastic modulus, all other selected second-phase inclusions either maintain or increase the effective modulus as their respective volume loading in Al_2_O_3_ matrix increases, which is associated with their intrinsic superior values of elastic modulus. SiC, cBN, and TiC tend to significantly increase the elastic modulus while loading of TiO_2_ and Si_3_N_4_ into Al_2_O_3_ do not tend to considerably alter the effective modulus of the composites when the volume fraction is increased, which is associated with comparable elastic modulus of the Al_2_O_3_ matrix. To quantify the relative effects of various second-phase inclusions, Figure 2b shows effective fracture toughness as functions of reinforcement volume fraction by assuming the particle size to be 40 µm in each case. Shape of inclusions are assumed perfectly spherical while the strength ratio is maintained constant. It can be observed that *K_IC_* increases with volume fraction but decreases as volume fraction goes beyond a peak value at around 20% for particles having low fracture resistance such as cBN and PSZ. However, this peak shifts towards the right when the fracture resistance of the inclusions increases and *K_IC_* keeps on increasing for particles such as SiC and TiO_2_ having very high fracture resistance values. Graphite is an exception here that tends to decrease the *K_IC_* even below the threshold value of 4.09 MPa·m^1/2^ when added in more than 30 vol % despite its high fracture resistance. This is due to its low stiffness and shear modulus values. The decrease in the *K_IC_* beyond certain volume fraction (in the case of reinforcements with low fracture energies) is associated with dominant particle cracking mode as compared to interface cracking at high volume fractions in the matrix. A similar phenomenon is also reported by Kumai et al. [34]. Hence, this trend suggests that crack deflection rather than particle cracking can be promoted and hence *K_IC_* can be enhanced with inclusions with high fracture energies for well-bonded interface as assumed in the current model. Hence, to get maximum enhancement in *K_IC_* of Al_2_O_3_ composite, particles such as SiC and TiO_2_ with high fracture energy can be used. Though graphite and TiO_2_ are considered excellent solid lubricants, loading of these particles in high volume fractions in Al_2_O_3_ matrix is not recommended due to their relatively low stiffness as evident from their intrinsic elastic moduli and effect on overall stiffness of the composite as shown in Figure 2a. However, a controlled amount of such particles in combination with other particles such as SiC can be used to get the composites, which will be discussed in the next section.

To quantify the influence of content and size of second-phase on the resulting *K_IC_* of the composites, cBN is considered as a second-phase inclusion in Al_2_O_3_ matrix with varying particle sizes as a function of volume fraction as shown in Figure 3a. A similar trend of increasing and then decreasing *K_IC_* is observed as a function of volume fraction for any particular size which indicates that particle-cracking tends to dominate the interface-debonding as volume fraction exceeds certain value. It is worth noting in Figure 3b that experimentally measured *K_IC_* values of sintered Al_2_O_3_ composites (with 10%, 20% and 30% volume of cBN) are in good agreement with the *K_IC_* predictions for a particle size of 40 µm. As far the effect of particle size on *K_IC_*, it can be noted that decreasing the particle size from 40 µm to 20 µm and then to 5 µm tend to increase *K_IC_* for any particular content of cBN. Moreover, the peaks of the curves corresponding to maximum *K_IC_* gradually shift toward the right as the particle size deceases from 40 µm to 5 µm. More precisely, the maximum value of *K_IC_* at 5 µm, 20 µm and 40 µm are 5.37 MPa·m^1/2^, 4.92 MPa·m^1/2^ and 4.56 MPa·m^1/2^ for volume fractions of 25%, 22.5% and 20%, respectively. This phenomenon of peak shifting again implies that embedding smaller particles in Al_2_O_3_ matrix tends to resist particle cracking for any particular volume fractions and hence results in higher value of *K_IC_* due to more crack deflection. In other words, the lower *K_IC_* values of composites at larger particle sizes embedded in Al_2_O_3_ matric are a result of an increased tendency of particle-cracking which is also reported by Evans [35] and Lin et al. [36]. Though crack penetration through a particle (resulting in cracking) requires a higher stiffness of particle and bulk modulus as compared to crack growth owing to the interfacial-debonding, particle-cracking typically shows rapid sudden failure of the composite. As far as smaller particles embedded in Al_2_O_3_ matrix are concerned, due to large surface area, crack deflections through interface between the matrix and particle tend to promote and hence result in enhanced *K_IC_*.

### 4.2. Hybrid Second-Phase Inclusions Mixed in Al_2_O_3_ Matrix

The previous single-inclusion study has led us to select the best combinations of second-phase particles that could be mixed in hybrid fashion to get the best functional properties of Al_2_O_3_ composite tool inserts. These functional properties are effective stiffness, fracture toughness together with self-lubricity attribute, which is required for applications in dry cutting operations.

The single inclusion studies have shown that the optimum candidates are to be SiC together with either cBN, TiO_2_ or graphite if mixed in appropriate proportions keeping in view their effect on the overall properties such as stiffness, fracture toughness and potential of using as solid lubricant. It is worth mentioning here that though graphite significantly reduces the stiffness and fracture toughness, its addition in small amount is expected to induce some tribological effects. In order to achieve the threshold elastic modulus of standalone Al_2_O_3_ and enhanced *K_IC_* together with expected low coefficient of friction and simultaneously maintaining high wear resistance, separate simulations are run where SiC is kept as a fundamental second-phase particle and cBN, TiO_2_ or graphite as an additional particle with varying volume fractions. It is worth mentioning here that during the experimental phase, a slight transformation of cubic boron nitride (cBN) to hexagonal boron nitride (hBN) has occurred (confirmed through Raman spectroscopy as will be discussed later) within the sintered Al_2_O_3_-cBN and Al_2_O_3_-SiC-cBN composites which is taken into account as one of the factors for deviation in computational and experimental results. Nevertheless, it is important to note that hBN is considered as one of the best candidates to be used as solid lubricant and hence using cBN as second-phase particle with favorable synthesis conditions for such transformation would be beneficial. After predicting the effective elastic modulus of Al_2_O_3_ composites with SiC and other second-phase, we have found that 20% cBN, 10% TiO_2_ or 5% graphite mixed with at least 20% of SiC in Al_2_O_3_ would give the best combinations in terms of intended properties. A large elastic modulus helps to resist the deformation of tool inserts occurring during machining. Figure 4 shows the predicted effective elastic modulus of Al_2_O_3_ composites as a function of volume fractions of SiC with 20% cBN, 10% TiO_2_ and 5% graphite as three different combinations. Al_2_O_3_ with SiC alone is also shown. For validation purposes, four composite systems namely Al_2_O_3_/20% SiC, Al_2_O_3_/20% SiC/20% cBN, Al_2_O_3_/20% SiC/10% TiO_2_ and Al_2_O_3_/20% SiC/5% graphite, are also highlighted in the figure which corresponds to the sintered composites developed through SPS process which will be discussed for validation purposes in the next section. It can be observed from Figure 4 that the effective modulus increases as the content of SiC increases with 20% cBN in Al_2_O_3_ matrix. The superior individual elastic modulus of both SiC and cBN is expected to result in an enhanced overall elastic modulus of the intended composite. Due to the very low average elastic modulus of graphite, it is not recommended to add a large amount of graphite in Al_2_O_3_ matrix though it is considered the best solid lubricant known. Hence, by keeping the amount of graphite as 5% in Al_2_O_3_ matrix, the effective modulus is predicted as a function of SiC volume fraction as shown in Figure 4. It can be seen that SiC volume fraction of at least ~23% is needed to maintain the threshold/benchmark elastic modulus of standalone Al_2_O_3_ (~280 MPa). However, there is no significant impact of TiO_2_ on the effective elastic modulus if used together with SiC for any volume fraction. Nevertheless, TiO_2_ is considered as an excellent solid lubricant and expected to enhance the effective fracture toughness as already discussed in the previous section. 

To improve the structural integrity of the proposed hybrid Al_2_O_3_ composites, it is necessary to improve the overall fracture toughness. It is evident that superior fracture toughness leads to restricted crack propagation, which is crucial for an effective cutting tool life. The effective fracture toughness is predicted for the proposed hybrid Al_2_O_3_/SiC/cBN, Al_2_O_3_/SiC/TiO_2_, and Al_2_O_3_/SiC/graphite composites as a function of volume fractions of both types of second-phase inclusions in each case. With Al_2_O_3_ as a base matrix and varying amount of SiC as the first inclusion, second inclusion, namely cBN, TiO_2_ and graphite, have been varied as shown in Figure 5a to Figure 5c, respectively. A particle size of 40 µm is assumed for all the inclusions. As can be seen in Figure 5a, adding cBN in Al_2_O_3_ matrix when the SiC volume content is less than ~10% is predicted to bring some enhancement in effective *K_IC_*. However, on the other hand, the overall *K_CI_* decreases for any content of cBN when SiC content is more than 10%. This is attributed to relatively low fracture resistance of cBN particles as compared to SiC, which tend to dominate particle cracking at cBN volume loading. These predictions also suggest that in order to enhance the overall *K_IC_* together with large content of cBN, a smaller particles size can be used. Figure 5b depicts that loading of TiO_2_ together with SiC can bring significant improvement in *K_IC_* of Al_2_O_3_ matrix composites. Yet, the effect of such increase gradually diminishes when the TiO_2_ addition reaches to a volume fraction of ~20%. At higher volume content, the *K_IC_* rather reduces if the SiC content becomes more than ~20% as depicted in the figure. A net increase in the *K_IC_* is associated with a high fracture resistance of TiO_2_ (68 J/m^2^) as compared to SiC (27 J/m^2^). Addition of TiO_2_ in excess amount i.e., beyond 20% tends to dominate the particle-cracking mode as discussed earlier. As shown in Figure 5c, due to extremely low stiffness and shear modulus, addition of graphite together with SiC reduces the *K_IC_* if compared with the case when only SiC is used with Al_2_O_3_. However, a little amount (5% to 10%) of graphite can induce significant tribological effects, which is desirable in dry cutting conditions. Based on the resulting *K_IC_* as depicted in Figure 5a–c, some combinations of composites with fixed amounts of cBN (20%), TiO_2_ (10%) and graphite (5%) as function of SiC are replotted in Figure 5d with some highlighted composites to be used for validation purposes (to be discussed in the next section).

### 4.3. Validation of Results

An experimental phase is performed to validate the accuracy of predictions made during the material design stage. This includes determination of properties and characterization of the composite samples. In order to validate the accuracy of the computational design, a series of Al_2_O_3_ matrix composites reinforced with single and hybrid second-phase inclusions are developed in line with predictions using spark plasma sintering. Effective properties such as elastic modulus and fracture toughness are measured experimentally and compared with the predictions. For accurate predictions, the base properties of sintered pure Al_2_O_3_ are first determined experimentally and then used in simulation. As mentioned earlier, both micro and nano-sized α-Al_2_O_3_ powders were used as matrix material for sintering the composites. The purpose of using two different sizes of Al_2_O_3_ is to investigate the transformation of cBN to hBN. Table 2 shows the predicted and experimentally measured values of elastic modulus and fracture toughness of series of composites developed. The porosity recorded in each sample is also shown. The deviation between the predicted and experimental values elastic modulus and fracture toughness is found to be less than 10%, which is considered to be in good agreement. There are some assumptions, which can be considered responsible for the discrepancy between the experimental and numerical results. These assumptions include spherical shape of inclusions, uniform distribution of particle size and perfect second-phase inclusion distribution within the matrix. More importantly, the inclusion data such as the values of fracture resistance and the interface fracture energy are taken from literature. It is important to note that interface debonding is promoted by spherical and smaller particle size, which could be held responsible for slightly higher predicted fracture toughness as compared to the measured values. This also suggests that irregular shape of particles as in the case of actual sintered composites are more prone to fewer complaint interfaces between the particles and matrix and hence promote particle cracking resulting in reduced fracture toughness. The elastic modulus of pure Al_2_O_3_ measured in the current work is 280 GPa which is associated with the sintering parameters and resulting porosity. The values of elastic modulus of sintered polycrystalline α-Al_2_O_3_ reported in the literature [23] are in the range of 221 GPa to 406 GPa which is shown to be dependent on the amount of porosity and other factors such as sintering type, relevant sintering parameters, and purity of Al_2_O_3_. The authors have previously reported [31] that porosity has a significant impact on the resulting stiffness and it is found that elastic modulus decreases with porosity nearly linearly.

### 4.4. Composites Characterization

The interaction between the second-phase inclusion(s) and Al_2_O_3_ matrix, the dispersion of second-phase particles as well as any phase transformation has direct bearing on the resulting properties such as stiffness and fracture toughness. A detailed microscopic analysis of the sintered sample is performed to characterize various composite combinations. Some representative FESEM images are shown in Figure 6a–d showing the pure Al_2_O_3_, Al_2_O_3_/20% cBN, Al_2_O_3_/20% SiC, and Al_2_O_3_/20% SiC/20% cBN, respectively. Some images are recorded on polished surfaces while others such as Figure 6d are from fractured surface. The composites with these compositions are also highlighted in Figure 4 and Figure 5d for reference purposes. The reinforcements of SiC and cBN can be seen as being embedded into the Al_2_O_3_ matrix. The presence of some ejected particles can also be noted which can be attributed to the brittle nature of the composite which leads to such ejection of particles. The porosity found can be related to the sintering parameters, and the composite density, which depends on the relative proportion of the matrix and second-phase particles.

To further confirm the presence of second-phase particles in the developed materials, Energy-Dispersive Spectrometry (EDS) analysis is performed for all composites to confirm the presence of each element present within the composite and the composite composition. A representative EDS analysis shown in Figure 7 is obtained from Al_2_O_3_-20% SiC-5% Graphite composite. It can be noted that the SiC is well-dispersed in the Al_2_O_3_ matrix while graphite dispersion is not up to the mark which is attributed very low percentage of graphite mixed in the composite.

XRD analysis is performed on the prepared samples to identify various phases in the resulting composites. Figure 8 shows XRD diffractogram recorded from surfaces of pure Al_2_O_3_, Al_2_O_3_/20% cBN, Al_2_O_3_/20% SiC/10% TiO_2_, and Al_2_O_3_/20% SiC/5% graphite. It is important to note that the XRD analysis of composites with micro-sized Al_2_O_3_ are presented here. The spectra located at various locations (diffraction angle (2θ) values) are compared with the standard values for Al_2_O_3_ and any other possible phases due to the reaction between Al_2_O_3_ and second-phase particles i.e., cBN, SiC, TiO_2_ and graphite. The XRD peaks reveal that no reaction occurred between Al_2_O_3_ and SiC, TiO_2_ and graphite and hence confirms only the presence of the constituent materials used in sintering the composites. However, XRD spectra corresponding to Al_2_O_3_/20% cBN has shown there is a slight amount of cBN phase that was transformed into hBN during the sintering process. It is believed that the thermo-mechanical stress during sintering were shifted to cBN particles thereby causing cBN to hBN phase transformation. It is previously reported [14,37] that such thermo-mechanical stresses are expected to arise due to uniaxial loading, plastic deformation during heating and the holding time, thermal shock, and mismatch of thermal expansion coefficient of the matrix and particles mixed in the composite.

To confirm the occurrence of cBN to hBN transformation phenomenon and dispel any concerns over any reactions between inclusions, Raman tests have been used to verify the XRD results and the presence of hBN. For this purpose, samples using nano-sized and micro-sized Al_2_O_3_ reinforced with 10%, 20% and 30% cBN each are sintered as single inclusion composites and Raman microscopic studies were performed on these samples. The FESEM images of Al_2_O_3_ composites can also be seen in Figure 3 as insets. The Raman spectra of cBN-reinforced Al_2_O_3_ composites (sintered from both micro and nano Al_2_O_3_ sizes) are shown in Figure 9a,b. The peaks at two different locations (at ~1050 cm^−1^ and ~1300 cm^−1)^, in both cases confirms the presence of cBN phase. However, an additional broad peak (at ~1360 cm^−1^) as in the case of micro Al_2_O_3_ can also be noted (Figure 9b) in addition to cBN peaks which is attributed to the transfer of thermo-mechanical stress according to the Al_2_O_3_ particle size. The absence of hBN transformation in the case of nano Al_2_O_3_ composite is due to enhanced surface energy due to large specific surface area associated with the nano particles, which tend to restrict such thermo- mechanical stress transfer. This tendency of transformation from cBN to hBN if sintered with micro Al_2_O_3_ matrix could be considered beneficial from tribological perspective keeping in view the fact that hBN is considered an excellent solid lubricant.

## 5. Conclusions

An innovative approach for the structural enhancement of Al_2_O_3_ based composites using computational tools is presented. Computational design codes based on mean-field homogenization and J-integral based fracture toughness models are used to predict the effective elastic modulus and fracture toughness of Al_2_O_3_ composites as a function of second-phase(s) attributes such as volume fraction, size and self-lubricity. The improved elastic modulus and fracture toughness is expected to help reduce the possibilities of failure due to shock and crack propagation. Numerous second-phase inclusions were considered and after selection of best candidates, computational simulations were used to select the optimum composite systems. The best combinations were then simulated as single and double inclusions in Al_2_O_3_ matrix composites and developed at the experimental phase using spark plasma sintering for means of validation. Various single and hybrid Al_2_O_3_ based composite systems were analyzed taking into consideration the effect of intrinsic properties and attributes of each second-phase particle on the overall properties of respective composite system. Fracture resistance and particle size of the second-phase inclusion were found the most significant factors in achieving the desired fracture toughness. cBN, TiO_2_ and graphite together with SiC mixed in Al_2_O_3_ matrix in appropriate combinations are found to have great potential to be used as toughened and self-lubricating composites. The desired properties such as elastic modulus and fracture toughness were measured experimentally to match the computational results. Results obtained from simulations were found to be in good agreement with the experimental data. It is expected that the proposed computational models can allow effective and economical composite design from a materials perspective for new ceramic tool inserts with various second-phase candidates.

## Figures and Tables

**Figure 1 materials-12-02378-f001:**
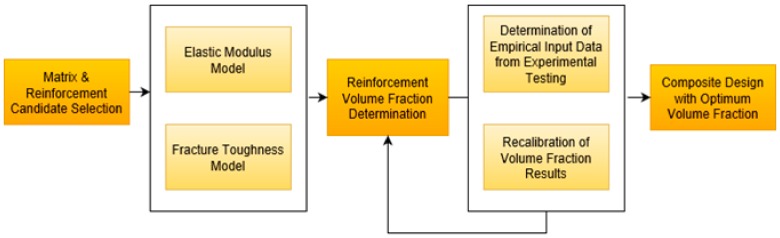
Flow chart showing steps used in material design stage.

**Figure 2 materials-12-02378-f002:**
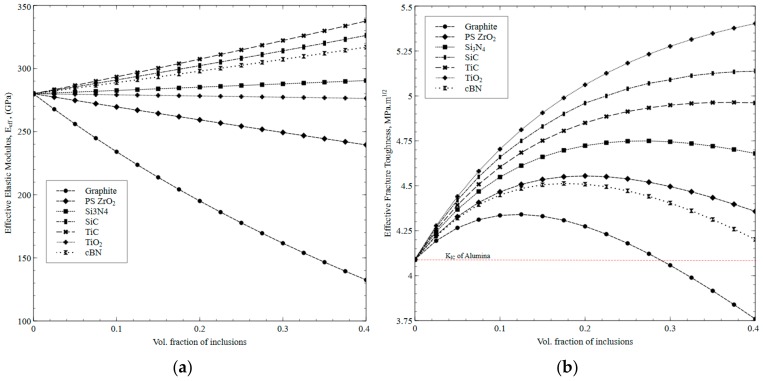
(**a**) Predicted effective elastic modulus, and (**b**) Predicted effective fracture toughness as a function of volume fraction of Al_2_O_3_ composite with the addition of independent inclusions.

**Figure 3 materials-12-02378-f003:**
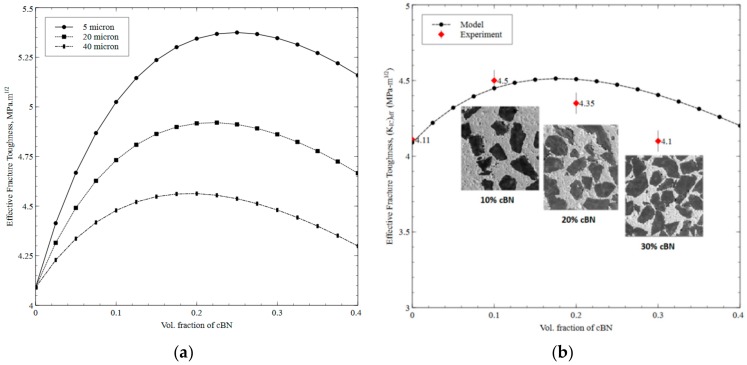
(**a**) Effect of particle size on effective fracture toughness of Al_2_O_3_ composite reinforced with cBN, (**b**) Comparison of model and experimental fracture toughness for varying volume fraction of cBN (40 µm) on resulting Al_2_O_3_ composite.

**Figure 4 materials-12-02378-f004:**
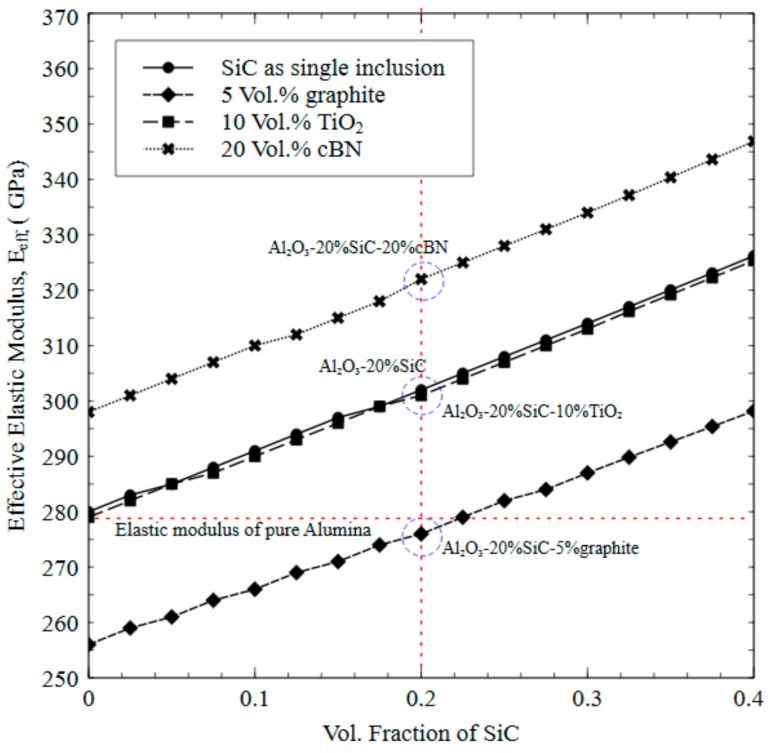
Predicted effective elastic modulus of Al_2_O_3_ composites as a function of volume fractions of SiC with 20% cBN, 10% TiO_2_ and 5% graphite. Al_2_O_3_ with SiC alone is also shown.

**Figure 5 materials-12-02378-f005:**
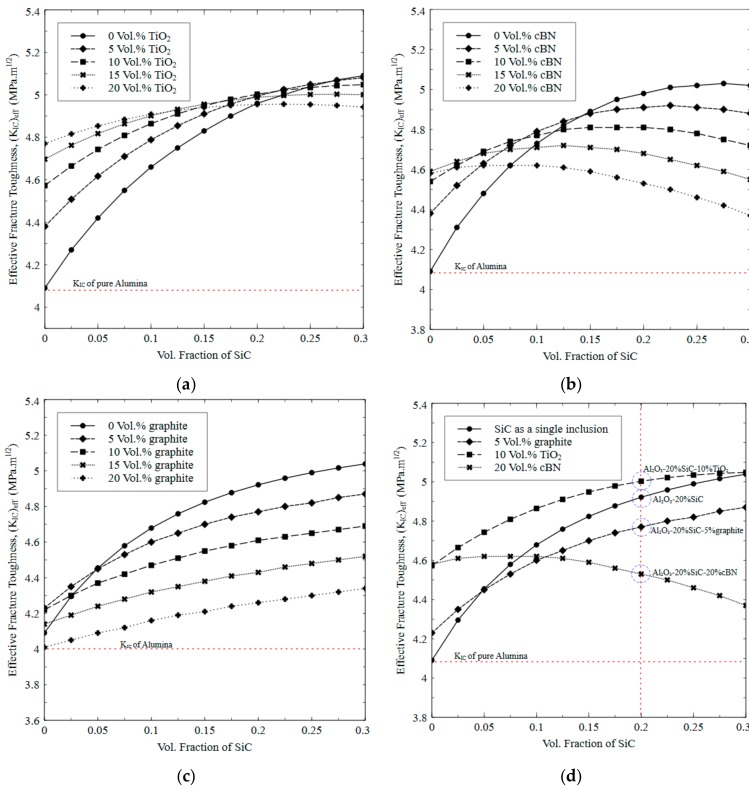
Predicted effective fracture toughness of Al_2_O_3_ composites as a function of SiC as a first inclusion and cBN, TiO_2_ and graphite as second inclusion. (**a**) Al_2_O_3_ with SiC and cBN, (**b**) Al_2_O_3_ with SiC and TiO_2_ and (**c**) Al_2_O_3_ with SiC and graphite, and (**d**) Predicted effective fracture toughness of Al_2_O_3_-SiC composites loaded with 20% cBN, 10% TiO_2_ and 5% graphite as a function of SiC volume content.

**Figure 6 materials-12-02378-f006:**
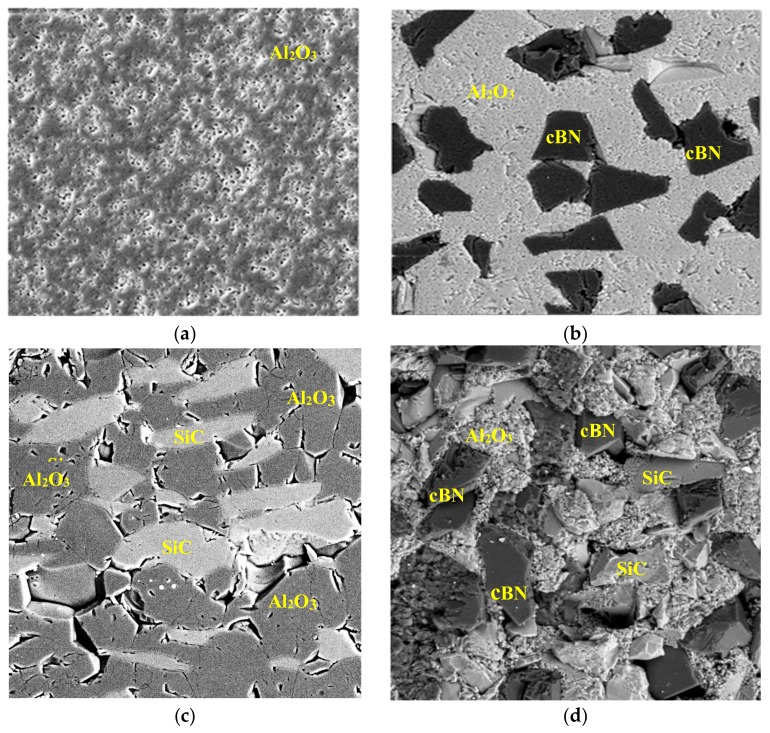
Field Emission Scanning Electron Microscope (FESEM) micrographs of Al_2_O_3_ matrix compositions: (**a**) pure Al_2_O_3_, (**b**) Al_2_O_3_/20% cBN, (**c**) Al_2_O_3_/20% SiC, (**d**) Al_2_O_3_/20% SiC/20% cBN.

**Figure 7 materials-12-02378-f007:**
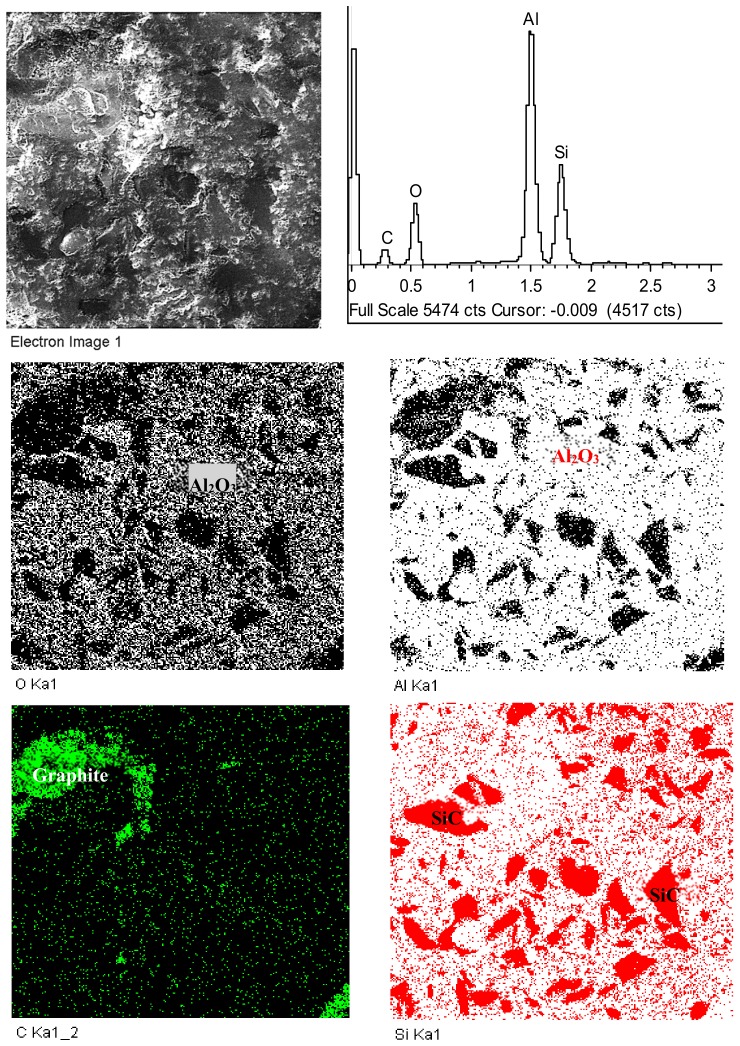
Energy-Dispersive Spectrometry (EDS) mapping highlighting the presence and location of each element within the Al_2_O_3_/20% SiC/5% Graphite composite.

**Figure 8 materials-12-02378-f008:**
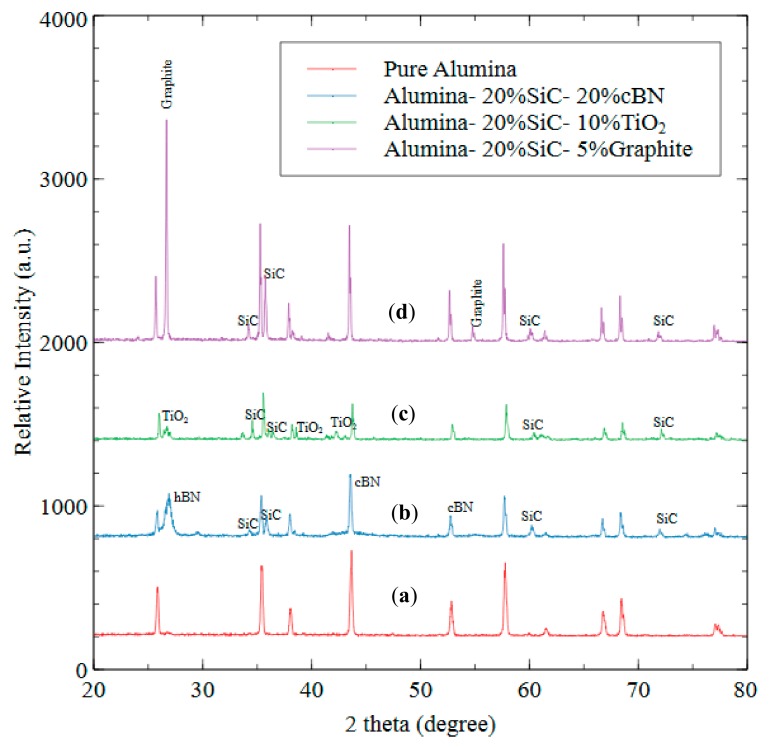
XRD pattern of (**a**) Pure Al_2_O_3_, (**b**) Al_2_O_3_/20% SiC/20% cBN, (**c**) Al_2_O_3_/20% SiC/10% TiO_2_, (**d**) Al_2_O_3_/20% SiC/5% Graphite.

**Figure 9 materials-12-02378-f009:**
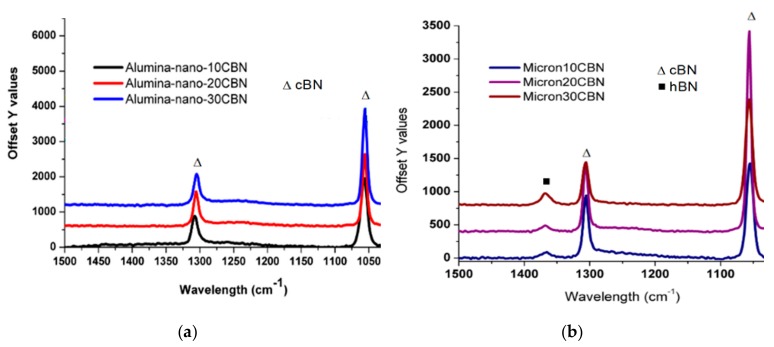
Raman spectra of (**a**) nano-sized Al_2_O_3_ composite reinforced with 10–30% cBN particles (**b**) micro-sized Al_2_O_3_ composite reinforced with 10–30% cBN particles.

**Table 1 materials-12-02378-t001:** Properties of Al_2_O_3_ and second-phase inclusions for Al_2_O_3_-based composite design used in predictions [23,32,33].

Candidates	Elastic Modulus (GPa)	Poisson’s Ratio	Bulk Modulus (GPa)	Shear Modulus (GPa)	Fracture Energy (J/m2)	Density (g/cm^3^)
Alumina (Al_2_O_3_)	280	0.22	165	124	54	3.96
Titanium Carbide (TiC)	450	0.18	250	152	45	4.94
Cubic Boron Nitride (cBN)	381	0.15	228	180	15	3.35
Silicon Nitride (Si_3_N_4_)	306	0.28	241	127	40.5	3.20
Titanium dioxide (TiO_2_)	270	0.28	210	112	68	4.09
Silicon Carbide (SiC)	410	0.14	250	180	27	3.1
Graphite	27.6	0.20	15.3	11.5	85	2.25
Partially Stabilized Zirconia (PSZ)	210	0.34	142	72	21	6.03

**Table 2 materials-12-02378-t002:** Comparison of experimental and predicted effective elastic modulus and effective fracture toughness of some representative Al_2_O_3_ composites.

Composition of Composite	Elastic Modulus, E (GPa)		Fracture Toughness, *K_IC_ (*MPa·m^1/2^)		Measured Porosity (%)
	Experimental	Predicted	Experimental	Predicted	
Pure Al_2_O_3_	280 ± 10	-	4.09 ± 0.35	-	2.37
Al_2_O_3_/10% SiC	305 ± 10	291	4.46 ± 0.20	4.66	2.63
Al_2_O_3_/20% SiC	312 ± 15	302	4.57 ± 0.10	4.96	2.69
Al_2_O_3/_30% SiC	321 ± 10	314	4.90 ± 0.20	5.09	2.76
Al_2_O_3_/10% cBN	301 ± 15	289	4.50 ± 0.40	4.45	3.10
Al_2_O_3_/20% cBN	313 ± 20	298	4.35 ± 0.30	4.50	3.26
Al_2_O_3_/30% cBN	306 ± 15	308	4.10 ± 0.25	4.40	3.28
Al_2_O_3_/20% SiC/20% cBN	328 ± 20	322	4.23 ± 0.20	4.53	3.43
Al_2_O_3_/20% SiC/10% TiO_2_	317 ± 15	301	4.55 ± 0.30	4.95	2.65
Al_2_O_3/_20% SiC/5% graphite	297 ± 15	276	4.62 ± 0.10	4.77	2.77

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
