# Peer review of "Design and Development of Hybrid Al_2_O_3_ Based Composites with Toughening and Self-Lubricating Second-Phase Inclusions"

_materials, 2019, doi:10.3390/ma12152378_

Round 1

Reviewer 1 Report

The present paper investigated an innovative approach for the structural enhancement of Al2O3 based composites. Computational design codes based on mean-field homogenization and J-integral based fracture toughness models are used to predict the effective elastic modulus and fracture toughness of Al2O3 composites as a function of second-phase attributes such as volume fraction, size and self-lubricity. Numerous second-phase inclusions were considered and a selection of best candidates was done using computational simulations. The best combinations are simulated as single and double inclusions in Al2O3 matrix composites and developed at the experimental phase using spark plasma sintering for means of validation.

Introduction is concise, systematized in order to guide the reader towards the objectives of the work. Innovation and objectives are clearly defined.

The mathematical model and experimental setup is described in detail with the appropriated references. However, in my opinion the text gained more clarity if the authors identified all the compositions developed (reinforcement of a single phase, dual phase, etc). Perhaps in a synthesis table.

Results and its discussion is careful and guides the reader through the optimization justifying the new steps.

Conclusion it concise and summarize very well the principal remarks done.

From my reading of the article I present some suggestions:

Page 2, line 48: the authors refer to thermal shock and mechanical shock, right?

Page 2, line 51-52: “ceramic-based tools becomes ductile at 2200oC in contrast to carbide tools, which soften at around the 870oC region” - This statement needs support, please insert reference!

Page 2, line 58-59: sentence confused? Please reformulate.

Page 2, line 62: “In recent published works…” Suggestion - insert 2 or 3 references.

Page 2, line 70: “is attributed to the enhanced hardness” - it is better toughness in place of hardness, no?

Page 2, line 71: “secondary phase [6,7],[8].” – it is better [6-8], please review and correct others cases along the manuscript.

Page 2, line 73: “Al2O3 reinforced with TiC significantly enhance” - how much? Quantify?

Page 2, line 74-76: “A ductile phase mixed into brittle ceramic composite is expected to improve fracture toughness, which is attributed to the dissipation of energy associated with crack propagation” - This statement needs support, please insert reference!

Page 2, line 76-77: “inclusion of cobalt and Zirconia inclusions into a Al2O3 reinforced with TiC possesses a reasonably higher toughness [12]” - how much? Quantify? Inclusion of cobalt and zirconia inclusions... suggestion - rephrase the sentence. Ref 12? In this reference the reinforcement was ceria not TiC. Also another comment - the ref's 10 and 11 were not mentioned. Follow the ascending order. Attention - ref 10 in reference list is repeated.

Page 2, line 78: “Li et al. [11] developed a novel” - Please confirm this reference!

Page 2, line 80: “Kumar et al [12]” – et al. (insert point), please review and correct others cases along the manuscript.

Page 3, line 99: “Al2O3” – attention to the indices, please review and correct others cases along the manuscript.

Page 3, line 139: “Eqn” – correct to “Equation” or “Eq.”

Page 3, line 144: “equation (1) and (2)” – equations?

Page 4, line 147-148: “There are two kinds of possible fracture types that can occur when a crack 147 approaches an interface between matrix and inclusion within a ceramic composite” - This statement needs support, please insert reference!

Page 4, line 167 – align the equation number.

Page 5, line 215: “where” – another line.

Page 5, line 222-225: sentence repeated? The main objective is describe above (page 3), in this point is unappropriated.

Page 5, line 232: “Some inclusions have been eliminated” - which ones?

Page 5, line 240: “Two types of α- Al2O3 powders with ~15 μm and ~150 nm sizes”- in alumina matrix the volume fraction of each powders sizes are similar?

Page 6, line 249: “disc shape with a dimeter of 20 mm” – diameter. Thickness?

Page 6, line 263-254: “to measure the fracture toughness by employing the mathematical relationship [29]” - are you sure about this reference?

Page 6, line 274: “reduced modulus (Er) is calculated using data from the indentation as” – add equation (13)

Page 7, Table 1: elastic modulus for alumina of 280 GPa - very low value - dense alpha alumina has E larger than 400GPa. Suggestion. Introduce more information about this alumina, e.g. theoretical porosity of the samples.

Page 7, line 304: “steps” – flow chart.

Page 7, line 315: “MPa-m1/2” - attention at units. It’s a point not a trace, e.g. “MPa.m1/2” - please review and correct others cases along the manuscript.

Page 8, line 359: “As far the effect of particle size on KIC, it can be noted that decreasing the particle size from 40 μm to 20 μm and then to 5 μm tend to decrease KIC” - The raw material of cBN has different size? they are milled?

Page 9, line 391: Figure 2 - suggestion: if the order of the caption follows the same order as the chart makes it easier to understand.

Page 10, line 426: “a slight transformation of cBN to hBN” – suggestion describe cubic to hexagonal.

Page 10, line 474 – throughout the text the authors refer tribological properties, but do not characterize the samples in tribological or wear tests. Suggestion: You should minimize these references.

Page 13, Figure 6 - large porosity is observed. It is possible quantify this?

Page 17, line 748 – repeated reference!

Reviewer 2 Report

Please find the comments in the attachments.

Reviewer 3 Report

The paper is interesting and well written.

Author Response

Thanks for the encouraging comments.

Round 2

Reviewer 2 Report

Comments are all addressed.